# Oscillation-Reduced MXFP4 Training for Vision Transformers

**Yuxiang Chen** [1 2]  **Haocheng Xi** [3]  **Jun Zhu** [1]  **Jianfei Chen** [1]

## Abstract

Pre-training Transformers in FP4 precision is becoming a promising approach to gain substantial speedup, but it comes with a considerable loss of accuracy. Microscaling (MX) data format provides a fine-grained per-group quantization method to improve the representation ability of the FP4 format and is supported by the next-generation Blackwell GPU architecture. However, training with MXFP4 data format still results in significant degradation and there is a lack of systematic research on the reason. In this work, we propose a novel training method TetraJet for a more accurate FP4 training. We comprehensively evaluate all of the quantizers involved in the training, and identify the weight oscillation problem in the forward pass as the main source of the degradation in MXFP4 training. Therefore, we introduce two novel methods, EMA Quantizer (Q-EMA) and Adaptive Ramping Optimizer (Q-Ramping), to resolve the oscillation problem. Extensive experiments on Vision Transformers demonstrate that TetraJet consistently outperforms the existing 4-bit training methods, and Q-EMA & Q-Ramping can provide additional enhancement by effectively reducing oscillation. We decreased the accuracy degradation by more than $50\%$ compared to the baseline, and can even achieve competitive performance compared to full precision training. The codes are available at https://github.com/thu-ml/TetraJet-MXFP4Training

## 1. Introduction

Low-precision training has emerged as a promising technique for accelerating the training process of large-scale neural networks. By quantizing tensors in both the forward and backward passes to lower-precision formats, low-precision training leverages specialized compute units in modern hardware to enhance computational efficiency significantly. While BF16 and FP16 precision remain the most widely used formats for deep learning training (Narang et al., 2017; Kalamkar et al., 2019), FP8 training (Sun et al., 2019; Micikevicius et al., 2022; NVIDIA, 2024c; Xi et al., 2025) is becoming increasingly mature in these years, with successful application in training state-of-the-art large language models (Liu et al., 2024).

There is a growing interest in pushing the training precision down to 4-bit. While earlier works attempt to train the network with FP4 (Sun et al., 2020), logarithm format (Chmiel et al., 2021), and INT4 (Xi et al., 2023), these works have rather large accuracy degradation (e.g., 1-2%) even on simple tasks such as ResNet training, and are not practically favorable. Recently, a Microscaling (MX) data format has been proposed for accurate low-precision training and inference (Rouhani et al., 2023b;a). MX applies fine-grained per-group quantization, where each small group of 32 elements shares a scaling factor. This fine-grained quantization scheme significantly mitigates the impact of outliers, and thus reduces quantization error. Particularly, the MXFP4 format utilizes an E2M1 (Exponent / Mantissa) FP4 with an E8M0 scaling factor. MXFP4 is supported on the latest Nvidia Blackwell architecture and is 2 times faster than FP8/MXFP6 and 4 times faster than FP16/BF16 (NVIDIA, 2024a;b) when doing matrix multiplications. However, the low-precision training method proposed in the original Microscaling paper uses MXFP6 activation/gradient, which is as slow as FP8 training. The fast pure MXFP4 training still has major accuracy degradation as tested in our experiments, which makes it infeasible to use in practice.

In this work, we propose **TetraJet**, a novel training method for transformer (Vaswani, 2017) with MXFP4 computation in both forward and backward pass. All weight/activation/-gradient tensors in linear layers are quantized to MXFP4 to fully unlock the acceleration potential of the hardware. We propose several techniques to improve the accuracy of

---
[1] Dept. of Comp. Sci. and Tech., Institute for AI, BNRist Center, THBI Lab, Tsinghua-Bosch Joint ML Center, Tsinghua University [2] Zhili College, Tsinghua University [3] University of California, Berkeley. Correspondence to: Jianfei Chen <jianfeic@tsinghua.edu.cn>.

*Proceedings of the 42nd International Conference on Machine Learning*, Vancouver, Canada. PMLR 267, 2025. Copyright 2025 by the author(s).

MXFP4 training. First, we propose a truncation-free scaling method for quantizing full-precision values to MXFP4 to avoid information loss in truncation. We further propose a double quantization method to deal with the non-square quantization group of MXFP4. With these techniques, we prove that TetraJet can estimate the gradient unbiasedly.

We then conduct a comprehensive evaluation of the impact of individual quantizers on the final model performance, and find that activation and weight quantizers in the forward pass contribute the most to accuracy degradation, due to a weight oscillation problem: the master weight fluctuates around the quantization boundary, causing the model to be quantized into different values across iterations, which consequentially brings significant instability in the optimization process. We propose two methods to alleviate the oscillation problem: the EMA quantizer (Q-EMA) conducts rounding based on the moving average of historical weights rather than only depending on the current weight matrix; and the Adapting Ramping optimizer (Q-Ramping) adaptively identifies and reduces the update frequency of oscillating weights.

Extensive experiments on Vision Transformers prove that TetraJet consistently outperforms Microscaling's original method (Rouhani et al., 2023b), and Q-EMA & Q-Ramping can provide additional improvement through oscillation reduction. We decreased the accuracy degradation by more than $50\%$ compared to the baseline, and even achieve competitive performance compared to full-precision training.

## 2. Related Work

**Low-Precision Training**  Low-precision training has become a prominent technique in modern deep learning to speed up the training process. FP16 and BF16 (half-precision) training (Narang et al., 2017; Kalamkar et al., 2019) is currently the most common low-precision method. FP8 and INT8 training (Sun et al., 2019; Zhu et al., 2020; Micikevicius et al., 2022; Wortsman et al., 2023; NVIDIA, 2024c; Peng et al., 2023; Xi et al., 2024; 2025; Liu et al., 2024) further improves efficiency and uses more fine-grained per-tensor / per-row / per-block quantization. When it comes down to 4-bit training (Sun et al., 2020; Chmiel et al., 2021; Xi et al., 2023), more techniques are being applied (e.g. Hadamard transformation) to compromise the degradation caused by the low representation ability. Still, their accuracy degradation is not negligible.

For a more fine-grained quantization, the Microscaling (MX) format (Rouhani et al., 2023a;b) in the Blackwell architecture (NVIDIA, 2024a) offers a $1 \times 32$ per-group quantization and could potentially double the speed compared to FP8 training. Rouhani et al. (2023b) also propose a low-precision training method with computation flow in MX formats in 4, 6, and 8 bits. In this paper, we refer to

the 4-bit MX format as *MXFP4*, and refer to their training method as *Microscaling*. We propose a better training method TetraJet with accuracy improvement compared to Microscaling.

**Oscillation Problem**  In low-precision training, weight oscillation has been proven to be a serious problem that affects optimization. Nagel et al. (2022) revealed that the oscillation of weight quantization does harm to Quantization-Aware Training (QAT) of CNNs. Besides, Liu et al. (2023) proved that oscillation was a key factor causing the degradation of accuracy in QAT of Vision Transformers. However, they were both based on QAT, that is, they *fine-tunes* a low-precision model based on a pre-trained full-precision network rather than *pre-training* from scratch. They both utilized pre-tensor Learned Step Size Quantization (LSQ) to train the models. There is a lack of research on oscillation problems about pre-training and more fine-grained quantization methods (e.g., MX Format).

To reduce weight oscillation, Liu et al. (2023) proposed several methods, but the application is restricted to LSQ or QAT, while Nagel et al. (2022) proposed methods that can be generalized to reduce oscillation in MXFP4 pre-training: The method "Dampen" tried to encourage latent weights to be closer to the quantized value to avoid fluctuating around the quantization boundary, by adding a regulation term $\mathcal{L}_{\mathrm{dampen}} = \|\mathbf{W} - Q(\mathbf{W})\|_F^2$ in the loss function; The method "Freeze" tracks the oscillation frequency $f$ for each weight element, and freezes those frequently oscillating weights ($f > f_{\mathrm{th}}$) to a running average value. The frozen weights would never be updated again in the whole training process, which may harm the optimization in pre-training. In this work, we propose two novel methods *Q-EMA & Q-Ramping* to better reduce oscillation in MXFP4 pre-training.

## 3. Our TetraJet Training Method

In this section, we review and identify several drawbacks of the existing low-precision training method **Microscaling** (Rouhani et al., 2023b), and propose a more accurate training method **TetraJet**. The effectiveness of our method is shown in Section 7.

### 3.1. Preliminary

**MXFP4 Format**  Floating points have three components: sign-bit, exponent-bits, and mantissa bits. If a format has $x$ exponent bits and $y$ mantissa bits, we usually denote it as E$x$M$y$. We use $Q_p, Q_n$ to represent the max positive value and the min negative value the format can represent. For E2M1, $Q_p = 6, Q_n = -6$.

The MXFP4 (Microscaling Floating-Point 4-bit) data format (Rouhani et al., 2023a) follows a per-group quantization

scheme where a group of $N = 32$ elements shares a common 8-bit exponential scaling factor $s$. Each element $X_i$ in the group is represented by $P_i$ in E2M1 format. The reconstruction of a floating-point value $X_i$ from its MXFP4 representation follows the formula:

$$X_i = P_i \times 2^s, \quad i = 1, 2, \ldots, 32$$

**Quantization** To quantize a matrix to MXFP4, we need to split it into blocks of size $1 \times 32$ (or $32 \times 1$), and then quantize each block to MXFP4. To quantize a block of 32 full-precision values $\{X_i\}_{i=1}^{32}$ to MXFP4, we first determine the E8M0 scale factor $S = 2^s$ with $|s| \le 127$. Each value $X_i$ is then mapped to a 4-bit FP4 representation $P_i$, such that:

$$P_i = \text{round}_{\text{FP4}}\left(\frac{X_i}{S}\right), \quad X_i \approx P_i \cdot S. \tag{1}$$

The quantized representation is stored as $\left(\{P_i\}_{i=1}^{32}, S\right)$, where $S$ is an 8-bit exponent, and $P_i$ is an FP4 value.

### 3.2. Quantization with Truncation-Free Scaling

**Computation of Scaling Factor** Microscaling computes the scale factor as follows

$$s = \lfloor \log_2 M \rfloor - E_{\max}, \quad S = 2^s, \tag{2}$$

where $M = \max_{1 \le i \le 32} |X_i|$ is the largest absolute value of the block, $E_{\max}$ represents the largest exponent in FP4 format[1]. A drawback of the approach is that the scaled value $X_i/S$ may fall outside the range $[Q_n, Q_p]$, and exceeding values will be truncated. For instance, if $M = 31$, the scaling factor will be $S = 2^s = 2^{4-2} = 4$. Since $M/S = 31/4 = 7.75$ exceeds the maximum representable value $Q_p = 6$, the value $M$ will be truncated to 6. Intuitively, large values carry more information, and such truncation will be harmful to retaining the precision of the network.

TetraJet equips a *truncation-free scaling* method:

$$s = \left\lceil \log_2 \frac{2\widetilde{M}}{Q_p - Q_n} \right\rceil, \quad S = 2^s,$$

where $\widetilde{M}$ equals to $M$ in most cases except when $M = 0$, we set $\widetilde{M}$ to a small number $\epsilon = 10^{-8}$ to avoid numerical issues. Compared to Eq. (2), we replace floor $\lfloor \cdot \rfloor$ with ceiling $\lceil \cdot \rceil$ to avoid truncation, and replace the numerical range from $[-2^{E_{\max}}, +2^{E_{\max}}]$ to a more accurate range $[Q_n, Q_p]$. In this way, $Q_n \le M/S \le Q_p$ always holds. For example, when $M = 31$, the scaling factor will be $S = 2^3 = 8$, so $M/S = 3.875$ still lies in the representation range of FP4.

---

[1]for E2M1, $E_{\max} = 2^{2-1} = 2$

**Deterministic & Stochastic Rounding of FP4 format** Now we discuss the $\text{round}_{\text{FP4}}(\cdot)$ operation in Eq. (1). With our scaling, all the values $X_i/S$ are in the range $[Q_n, Q_p]$. Therefore, we can always find two consecutive FP4 value $q_1, q_2 (q_1 < q_2)$ satisfying $q_1 \le X_i/S \le q_2$ for every $X_i$.

A direct way of rounding $X_i/S$ is to select the nearest FP4 value between $q_1, q_2$, which we call *deterministic rounding* or *round to nearest*. Here we denote it as

$$\text{round}_D(X_i/S) = \begin{cases} q_1, & |X_i/S - q_1| < |X_i/S - q_2| \\ q_2, & \text{otherwise} \end{cases}$$

Microscaling always applies deterministic quantization to minimize the quantization error. However, we find it suboptimal to apply deterministic quantization to gradients, since the gradient will no longer be unbiased. To this end, we apply *stochastic rounding* (Courbariaux et al., 2015) to gradients to maintain an unbiased gradient. Stochastic rounding generates random variable $\xi \sim \text{Uniform}(-\frac{q_2-q_1}{2}, \frac{q_2-q_1}{2})$ for each value $X_i$ independently, and computes

$$\text{round}_S(X_i/S) = \begin{cases} q_1, & X_i/S + \xi < \frac{q_1+q_2}{2} \\ q_2, & \text{otherwise} \end{cases}$$

Stochastic rounding is unbiased: $\mathbb{E}[\text{round}_S(X_i/S)] = X_i/S$. We show the superiority of stochastic rounding in the ablation study in Sec. 7.3,

### 3.3. TetraJet Linear Layer

When training the transformer, linear layers usually take most of the computation. Following previous works on low-precision training (Xi et al., 2023), we mainly focus on accelerating the linear layer with MXFP4, whose forward and backward pass are defined as:

$$\mathbf{Y} = \mathbf{X}\mathbf{W}^\top,$$
$$\nabla_{\mathbf{X}}\mathcal{L} = (\nabla_{\mathbf{Y}}\mathcal{L})\mathbf{W}, \quad \nabla_{\mathbf{W}}\mathcal{L} = (\nabla_{\mathbf{Y}}\mathcal{L})^\top \mathbf{X},$$

where $\mathbf{X} \in \mathbb{R}^{N \times D}, \mathbf{W} \in \mathbb{R}^{C \times D}, \mathbf{Y} \in \mathbb{R}^{N \times C}, \mathcal{L}$ is a loss function, and $\nabla_{\mathbf{X}}\mathcal{L}/\nabla_{\mathbf{Y}}\mathcal{L}/\nabla_{\mathbf{W}}\mathcal{L}$ are the input/output/weight gradient matrices with the same size of $\mathbf{X}, \mathbf{Y}, \mathbf{W}$.

To accelerate training, we need to compute all three matrix multiplications (MMs) in MXFP4. To achieve this, we need to quantize the six input matrices of the three MMs to MXFP4, which can be formulated as:

$$\mathbf{Y} = Q_D^{(1)}(\mathbf{X}) \times Q_D^{(2)}(\mathbf{W}^\top) \tag{3}$$

$$\nabla_{\mathbf{X}}\mathcal{L} = Q_S^{(3)}(\nabla_{\mathbf{Y}}\mathcal{L}) \quad \times Q_S^{(4)}\left(Q_D^{(2)}(\mathbf{W}^\top)^\top\right) \tag{4}$$

$$\nabla_{\mathbf{W}}\mathcal{L} = Q_S^{(5)}\left((\nabla_{\mathbf{Y}}\mathcal{L})^\top\right) \times Q_S^{(6)}\left(Q_D^{(1)}(\mathbf{X})\right) \tag{5}$$

where $Q_D/Q_S$ refers to the deterministic/stochastic rounding quantizer. We explain the design of TetraJet linear layer as follows.

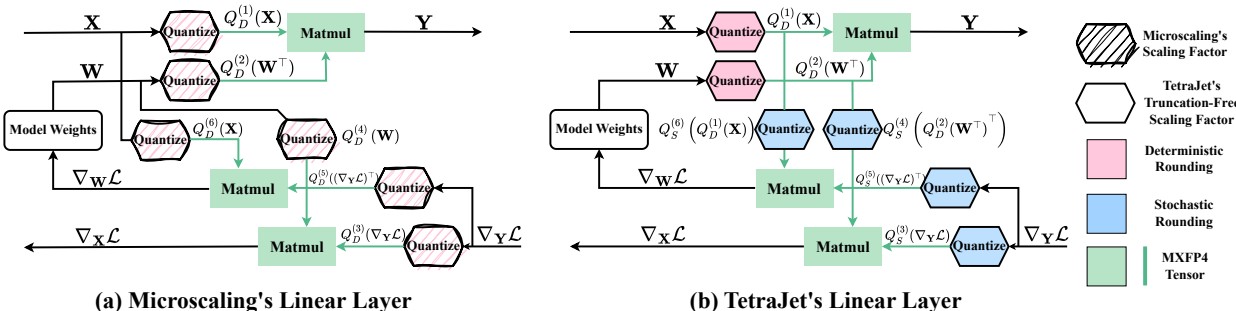

Figure 1: Visualization of MXFP4 Linear Layer.

**Block Format** As a fine-grained format, doing MM with MXFP4 is more subtle than other coarser-grained formats such as per-tensor quantization. For hardware-accelerated MM to be possible, MXFP4 requires quantization group shape to be $1 \times 32$ for the first matrix and $32 \times 1$ for the second matrix. Therefore, quantizers $Q^{(1)}, Q^{(3)}, Q^{(5)}$ should use $1 \times 32$ group shape, and quantizers $Q^{(2)}, Q^{(4)}, Q^{(6)}$ should use $32 \times 1$ group shape. This means that weight $\mathbf{W}$, activation $\mathbf{X}$, and gradient $\nabla_{\mathbf{Y}} \mathcal{L}$ should be quantized along different axes in different quantizers. For example, the quantization block size of $\mathbf{X}$ should be 1 token $\times$ 32 channels in forward and 32 tokens $\times$ 1 channel in backward.

**Double Quantization** We propose a *double quantization* strategy to satisfy MXFP4's block format requirement. Specifically, $Q_D^{(1)}(\mathbf{X})$ is a quantized activation with $1 \times 32$ group size, which is used in the forward pass. We quantize the already quantized $Q_D^{(1)}(\mathbf{X})$ *again* with a different $32 \times 1$ group size to compute the gradient in Eq. (5). By doing so, we ensure the activation is quantized with the required group size for both forward and backward pass. Similarly, the weight is also doubly quantized.

In contrast, Microscaling takes a different approach:

$$\nabla_{\mathbf{X}} \mathcal{L} = Q_D^{(3)}(\nabla_{\mathbf{Y}} \mathcal{L}) \times Q_D^{(4)}(\mathbf{W}) \tag{6}$$

$$\nabla_{\mathbf{W}} \mathcal{L} = Q_D^{(5)}\left((\nabla_{\mathbf{Y}} \mathcal{L})^\top\right) \times Q_D^{(6)}(\mathbf{X}) \tag{7}$$

where the activation used in the backward pass is *deterministically* quantized from the *full-precision* $\mathbf{X}$ rather than $Q_D^{(1)}(\mathbf{X})$, which is biased as we will discuss.

### 3.4. Gradient Bias

We first derive the correct gradient formula with Straight Through Estimator (STE) (Bengio et al., 2013): Given the forward pass in Eq. (3), the correct gradient should be

$$\nabla_{\mathbf{X}} \mathcal{L} \overset{\text{STE}}{\approx} \nabla_{Q_D^{(1)}(\mathbf{X})} \mathcal{L} = (\nabla_{\mathbf{Y}} \mathcal{L}) \times Q_D^{(2)}(\mathbf{W}^\top)^\top \tag{8}$$

$$\nabla_{\mathbf{W}} \mathcal{L} \overset{\text{STE}}{\approx} \nabla_{Q_D^{(2)}(\mathbf{W})} \mathcal{L} = (\nabla_{\mathbf{Y}} \mathcal{L})^\top \times Q_D^{(1)}(\mathbf{X}). \tag{9}$$

Table 1: Impact analysis on MXFP4 quantizers. We report the top-1 Acc.% after 90-epoch pre-training. Q$i$ means we only activate the $i$-th quantizer $Q^{(i)}$.

|  | DeiT-T | DeiT-S |
|---|---|---|
| Full Precision | 63.73 | 73.33 |
| Q1 | 61.50 | 71.66 |
| Q2 | 62.77 | 72.45 |
| Q3 | 63.46 | 72.97 |
| Q4 | 63.37 | 72.79 |
| Q5 | 63.81 | 73.25 |
| Q6 | 63.78 | 73.13 |
| All Quantizers | 59.75 | 71.03 |

Note that microscaling's gradient Eq. (6,7) does not equal to the correct gradient Eq. (8,9). Particularly, $Q_D^{(4)}(\mathbf{W}) \neq Q_D^{(2)}(\mathbf{W}^\top)^\top$. Microscaling is actually computing the gradient for *another network* with the forward pass $\mathbf{Y} = Q_{32 \times 1}(\mathbf{X}) Q_{1 \times 32}(\mathbf{W}^\top)$, where both operands are quantized in the wrong direction.

In contrast, TetraJet gives an unbiased estimation of Eq. (8,9). Take $\nabla_{\mathbf{X}} \mathcal{L}$ in Eq. (4) as an example, since $Q^{(3)}, Q^{(4)}$ are stochastic and truncation-free, the expectation of our gradient is

$$\mathbb{E}\left[Q_S^{(3)}(\nabla_{\mathbf{Y}} \mathcal{L}) \times Q_S^{(4)}\left(Q_D^{(2)}(\mathbf{W}^\top)^\top\right)\right]$$
$$= \mathbb{E}\left[Q_S^{(3)}(\nabla_{\mathbf{Y}} \mathcal{L})\right] \times \mathbb{E}\left[Q_S^{(4)}\left(Q_D^{(2)}(\mathbf{W}^\top)^\top\right)\right]$$
$$= \nabla_{\mathbf{Y}} \mathcal{L} \times Q_D^{(2)}(\mathbf{W}^\top)^\top$$

which is right side of Eq. (8). Similarly, the estimation in Eq. (5) for $\nabla_{\mathbf{W}} \mathcal{L}$ is also unbiased. Given that each linear layer is unbiased, the final gradient calculated with backpropagation is unbiased, which ensures the convergence of SGD, as discussed by (Chen et al., 2020).

### 3.5. Impact Analysis of Six Quantizers

Before making any attempts to improve the training, it is necessary to understand which among the 6 quantizers in

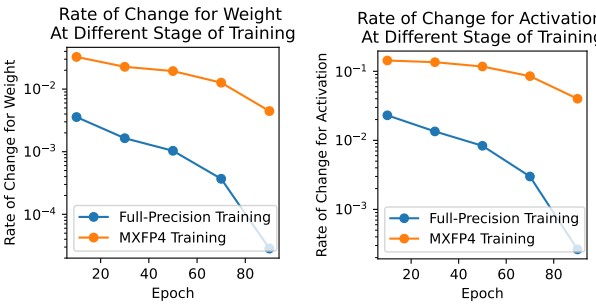

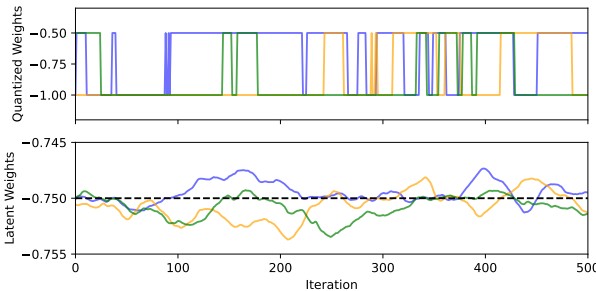

Figure 2: Rate of change for weight and activation at different stages of 90-epoch DeiT-Tiny pre-training. We calculate the average rate for all quantized weights and select a transformer block to test output activation given fixed input.

Figure 3: Trajectory of some oscillation elements in DeiT-Tiny during the last epoch of training. The top plot shows the change of quantized FP4 value, and the bottom plot shows the oscillating latent weight around the quantization decision threshold $\mathrm{thrd} = -0.75$.

Eq. (3,4,5) is the bottleneck. We test the impact of quantizers by activating them separately: for the $i$-th test, we only activate $Q^{(i)}$ while leaving all other matrices in full precision, train the model from scratch, and compute validation accuracy. As shown in Tab. 1, the activation/weight quantizers $Q^{(1)}/Q^{(2)}$ in the forward pass lead to most accuracy degradation. For example, MXFP4 training on DeiT-T has a 3.98% accuracy loss, while only quantizing the activation/weight in the forward pass accounts for 2.23% / 0.96%, respectively. We reveal in the next section this is due to the instability of low-precision training.

## 4. Oscillation Phenomenon

### 4.1. Instability of MXFP4 Training

During the final stage of training, the learning-rate (LR) typically approaches zero, so the model can stop exploration and quickly descend to a local minimum. However, we find that MXFP4 training *cannot converge* even with a sufficiently small learning rate due to the *oscillation* between quantization points. To explain this phenomenon, we define *rate of change* for a tensor $\mathbf{X}$ as

$$r(\mathbf{X}) = \frac{1}{T_0} \sum_{t=1}^{T_0} \frac{\left\| \mathbf{X}^t - \mathbf{X}^{t-1} \right\|_F}{\left\| \mathbf{X}^{t-1} \right\|_F},$$

where $t$ refers to training step, and step $0 \sim T_0$ refers to a short training interval. During pre-training, we can test the rate of change for the master weight $\mathbf{W}$, the quantized weight matrix $Q^{(2)}(\mathbf{W}^\top)^\top$, and activation $\mathbf{Y}$ at different stages.

As shown in Fig. 2, for full-precision models, the rate of change can gradually decrease to near zero, while for quantized models the rate of change would stay high in the final of training, indicating that there are still large changes inside the models.

We find that the *weight oscillation* is the source of this problem. To be clear, we refer to $w/S$ as *latent weight*, where $S$ is the quantization scale factor of weight element $w$. As illustrated in the top plot in Fig. 4, a large amount of latent weights lies around the quantization thresholds (the midpoints of two quantized values) at the end of the training process. For these elements, little perturbation on their corresponding master weights will change the quantized values, which results in a giant jump from one quantized value to another. This makes the rate of change of the quantized weight matrix much higher than its corresponding master weight, and meanwhile contributes to the instability of activation, which aligns with our finding.

We tracked several oscillating weight elements during the final epoch of training for a better understanding of this oscillation phenomenon. As shown in Fig. 3, these latent weights are changing with small steps around the *quantization threshold* $\mathrm{thrd} = -0.75$, which is the midpoint of two FP4 values $q_1 = -1, q_2 = -0.5$. When the latent weight crosses $\mathrm{thrd} = -0.75$ caused by a small update, the quantized weight would shift from $q_1$ to $q_2$ (or from $q_2$ to $q_1$). Frequently crossing $\mathrm{thrd}$ causes the frequent flipping between $q_1$ and $q_2$. Therefore, a direct characterization of oscillating weight is that, the oscillating weight elements will have their latent value stay closely around the quantization threshold and frequently cross the threshold.

### 4.2. Quantization Confidence of Weight Distribution

To quantitatively assess the severity of the oscillation problem, we define *quantization confidence* for each weight element $w$, which measures the normalized distance to the nearest quantization threshold:

$$\mathrm{QuantConf}(w) := \frac{\min_i |w - \mathrm{thrd}_i|}{\mathrm{MaxDist}(w^{\mathrm{FP4}})},$$

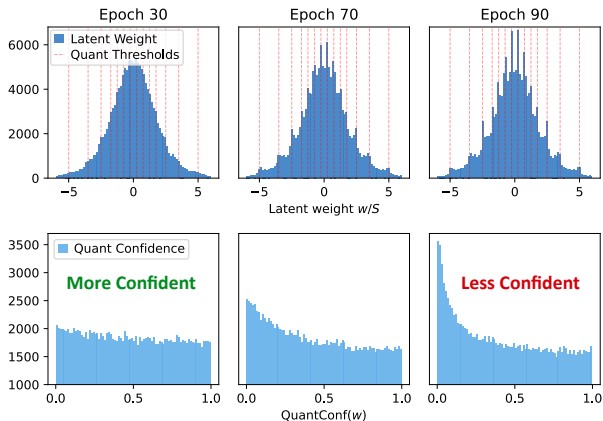

Figure 4: The change of latent weight and quantization confidence during 90-epoch pre-training of DeiT-Tiny. The top plot shows the distribution of latent weight, and the bottom plot shows the distribution of quantization confidence.

where $w^{\text{FP4}}$ denotes the quantized FP4 value of $w$, $\{\text{thrd}_i\}$ denotes all the quantization thresholds, and $\text{MaxDist}(w^{\text{FP4}})$ denotes the maximum possible distance if quantized to $w^{\text{FP4}}$. It is ensured that $\text{QuantConf}(w) \in [0, 1]$.

The rationale behind this metric is that the closer a latent weight is to a quantization decision threshold, the more likely it is to oscillate, making it harder for the weight to converge to a stable FP4 value.

As shown in the bottom plot of Fig. 4, we observe a gradual decline in quantization confidence throughout training. This trend indicates an increasing prevalence of oscillation as training progresses. Consequently, effective solutions to mitigate oscillation should be dynamic, adapting to the specific conditions of each stage of the training process.

## 5. EMA Quantizer

We firstly propose an *EMA Quantizer* (**Q-EMA**) to solve the oscillation phenomenon. Since the weight will oscillate between the two possible choices randomly even with small perturbations, we hope to find a better way to choose from these two possible values after quantization.

We find that the Exponential Moving Average (EMA) can be used to alleviate the oscillation problem. EMA on weight is determined as:

$$\mathbf{W}_{\text{EMA}}^t = \beta \mathbf{W}_{\text{EMA}}^{t-1} + (1 - \beta)\mathbf{W}^t, \qquad (10)$$

where $\mathbf{W}^t$ is the BF16 weight. A Typical choice of $\beta$ is 0.998. Therefore, even when weight makes a very large step, $\mathbf{W}_{\text{EMA}}$ only moves slightly. When the weight oscillates between two quantized values, as EMA weight is always left

behind the actual optimization process and is updated slowly, EMA weight is less likely to be affected by oscillations. Consequently, this makes the optimization process much more stable.

Our EMA quantizer first maintains an EMA weight throughout the training process. When doing quantization to each weight element $w$ with scale factor $S$ and its EMA value $w_{\text{EMA}}$, we first use the latent weight $w/S$ to propose two candidate quantized values $w_{q_1}$ and $w_{q_2}$, as they are the two values that give the smallest MSE. We then use the EMA weight to check which is closer to $w_{\text{EMA}}$, and use this as the quantized value. This algorithm is formalized as Algorithm 1 in Appendix C.

## 6. Adaptive Ramping Optimizer

Besides smoothing the weight quantization with EMA quantizer, another effective approach to reducing oscillations is to manually decrease the update frequency of oscillating weights. Building on this idea, we propose *Adaptive Ramping Optimizer* (**Q-Ramping**), which directly locates the frequently oscillating weights according to their updating trajectory, and then adaptively decrease their updating frequency by using a higher gradient accumulation step for these oscillating weights to reduce the oscillation frequency.

### 6.1. Identifying Oscillating Weights

The first thing is to locate the oscillating weights and quantify their degree of oscillation. To achieve this, we would record information about the weight update trajectory for each element. During a training stage with $T_0$ steps, we sum up updating distance for each master weight element $w$ and its quantized weight $w_Q$:

$$\text{dist}_W = \sum_{t=1}^{T_0} |w^t - w^{t-1}|, \quad \text{dist}_Q = \sum_{t=1}^{T_0} |w_Q^t - w_Q^{t-1}|,$$

And then, we define *oscillation ratio* $R_w$ for each weight element as

$$R_w := \text{dist}_Q / \text{dist}_W,$$

representing the degree of oscillation.

During training, if a weight element $w$ doesn't fall into the oscillation process, the master weight $w$ and the quantized weight $w_Q$ would move with a similar trajectory. In this situation, $\text{dist}_Q \approx \text{dist}_W$, so $R_w$ would not be too large.

In contrast, for oscillating weight elements, the quantized weight would switch frequently between two discrete quantization values $q_1$ and $q_2$. Each switch from $q_1$ to $q_2$ (or from $q_2$ to $q_1$) will increase $\text{dist}_Q$ by $|q_1 - q_2|$, making it relatively large. Meanwhile, the master weight $w$ would be oscillating around the quantization threshold, and the step-

Table 2: Results on the 90-epoch pretraining of Vision Transformers. We report the Top-1 Accuracy% on validation dataset.

| Pre-Training Methods | Bit Width | Quantization | DeiT-T | DeiT-S | DeiT-B | Swin-T | Swin-S |
|---|---|---|---|---|---|---|---|
| Full Precision | A16W16G16 | - | 63.73 | 73.33 | 75.57 | 78.35 | 80.44 |
| INT4 | A4W4G4 | Per-Tensor | 40.14 | 60.07 | 68.13 | 74.22 | 75.74 |
| MicroScaling (Baseline) | A4W4G4 | Per-Group | 58.56 | 70.10 | 74.54 | 76.87 | 79.45 |
| TetraJet (Ours) | A4W4G4 | Per-Group | 59.75 | 71.03 | 74.91 | 77.12 | 79.51 |
| TetraJet + Q-EMA(Ours) | A4W4G4 | Per-Group | 60.00 | **72.25** | **77.32** | 77.30 | **79.74** |
| TetraJet + Q-Ramping(Ours) | A4W4G4 | Per-Group | **60.31** | 71.32 | 75.62 | **77.33** | 79.67 |

size would be $\ll |q_1 - q_2|$, so in this situation, we would get $\mathrm{dist}_W \ll \mathrm{dist}_Q$, and $R_w$ will be quite large.

Therefore, the larger $R_w$, the more frequently and severely the weight element $w$ oscillates, which means that we should put more effort into suppressing the oscillation of $w$.

### 6.2. Suppressing Weight Oscillation Adaptively

Based on periodically detecting and quantifying weight oscillation, we propose *Adaptive Ramping Optimizer* (Q-Ramping) to alleviate the oscillation problem of these weights. We adaptively decrease the updating frequency of these oscillating weights, by setting larger batch-size for them. We also expand their corresponding learning-rate proportional to their batch-size. The adapted batch-size would be an integer multiple of the global batch-size, and we would accumulate the gradient for each oscillating weight according to its own batch-size. This algorithm can be formalized as Algorithm 2 in Appendix C.

By applying Q-Ramping, the update frequency is reduced for oscillating weights, so that their oscillation frequency is also reduced. Additionally, through larger batch-size and larger learning-rate, the oscillating weights near the quantization thresholds can be updated to a place further away from the quantization threshold. Therefore, the weight distribution will have a higher quantization confidence, and the oscillation phenomenon can be alleviated.

## 7. Experiments

### 7.1. Vision Transformer Pre-Training

We evaluate our TetraJet training method and oscillation reduction method Q-EMA & Q-Ramping on Vision Transformers pre-training. During training, we quantize the forward and backward process of all the linear layers in the Attention module and the MLP module of transformer blocks.

We do pre-training for DeiT-Tiny, DeiT-Small, and DeiT-Base (Touvron et al., 2021) using Facebook's training recipe[2], and pre-train Swin-Tiny and Swin-Small (Liu et al.,

2021) based on the official implementation[3]. All the models are trained for 90 epochs on ImageNet1K (Russakovsky et al., 2015) with default training recipes. For Q-EMA & Q-Ramping, we show the insensitivity to their hyperparameter choice in Appendix C.3.

We compared our MXFP4 training method, TetraJet, with full-precision training, 4-bit per-tensor quantization method INT4 (Xi et al., 2023), and original Microscaling's MXFP4 training method (Rouhani et al., 2023b). The detailed results are listed in Tab. 2.

As a result, our TetraJet can consistently outperform the original method Microscaling, and we can further improve the performance of MXFP4 training by overcoming oscillation problems in forward pass with Q-EMA / Q-Ramping.

### 7.2. Quantitative Analysis on Oscillation Reduction

To validate our improvements in mitigating oscillation, we analyzed different statistics to show how our methods work in oscillation reduction in real training.

**Improvement of Training Stability** As described in Sec. 4.1, the *rate of change* for weights and activation cannot converge to zero in MXFP4, which reflects the model cannot converge stably. In Tab. 3, we can see our methods can effectively reduce the instability of both the weight and activation in forward.

**Improvement of Quantization Confidence** As described in Sec. 4.2, *weight confidence* indicates the risk of weight oscillation. If the confidence is lower at the end of the training, more weights are still oscillating around the quantization threshold, and it is harder for these parameters to converge to decisive values.

In Fig. 5, we can see the unique function of Q-Ramping in improving quantization confidence. It successfully reduced the weights that are prone to oscillate (those with low confidence) by identifying them, reducing their update frequency, and increasing their gradient accumulation steps.

---

[2]https://github.com/facebookresearch/deit

[3]https://github.com/microsoft/Swin-Transformer

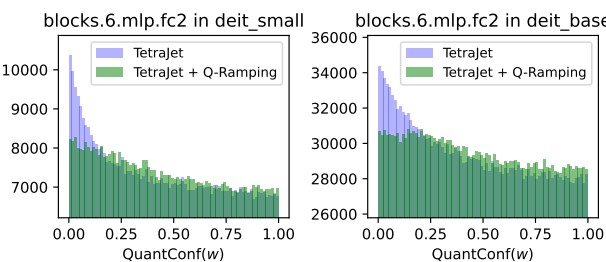

Figure 5: Q-Ramping's unique effect on improving the distribution of quantization confidence of the final model.

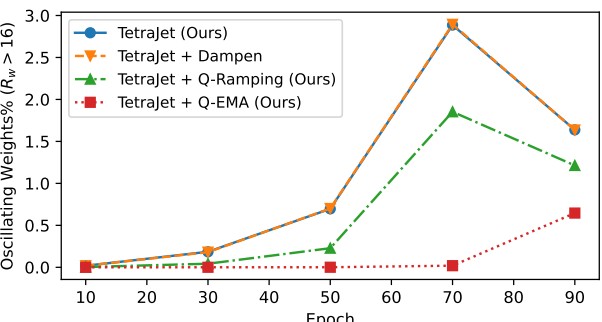

Figure 6: Q-EMA & Q-Ramping's effect on oscillating weights reduction during the whole training process. We present the 90-epoch pre-training of DeiT-T.

Table 3: Effect of Q-EMA & Q-Ramping on stabilizing weight and activation at the end of DeiT-T training. $r(\cdot)$ refers to the rate of change for tensors, $\mathbf{W}^Q$ is the quantized weights, and $\mathbf{Y}$ is the output of 9th transformer block given fixed input.

|  | $r(\mathbf{W}^Q)\downarrow$ | $r(\mathbf{Y})\downarrow$ |
|---|---|---|
| TetraJet | 0.0045 | 0.0401 |
| TetraJet + Dampen | 0.0044 | 0.0394 |
| TetraJet + Q-EMA (Ours) | 0.0018 | 0.0251 |
| TetraJet + Q-Ramping (Ours) | 0.0028 | 0.0318 |

Table 4: Comparison of our oscillation reduction methods with other methods for DeiT MXFP4-Pretraining on ImageNet Classification. We report the top-1 Acc.% of the final model.

|  | DeiT-T | DeiT-S |
|---|---|---|
| TetraJet | 59.75 | 71.03 |
| TetraJet + Dampen | 59.75 | 70.75 |
| TetraJet + Freeze | 16.45 | 22.04 |
| TetraJet + Q-EMA (Ours) | 60.00 | **72.25** |
| TetraJet + Q-Ramping (Ours) | **60.31** | 71.32 |

**Oscillation Reduction throughout the Training**  We use *Oscillation Ratio $R_w$* (defined in Sec. 6.1) to characterize the oscillation problem during the whole training process. We define that those weights with $R_w > 16$ are oscillating weights. As shown in Fig. 6, both of our methods can effectively reduce the Oscillating Weights. Among them, Q-EMA reduces the most oscillating weights by directly smoothing weight quantization. Q-Ramping also reduces the oscillating level, while method "Dampen" from Nagel et al. (2022) cannot effectively reduce oscillation in MXFP4 pre-training.

### 7.3. Ablation Study

**Training Method**  We investigate the quantization method in the MXFP4 training. We find that *double quantization* consistently outperforms Microscaling's incorrect gradient computation. Besides, when we ensure unbiased gradient estimation by *double quantization* and *truncation-free scaling*, we can get the optimal result with *stochastic rounding*. The detailed results are listed in Tab. 5 in Appendix B.

**Other Methods on Oscillation Reduction**  Following the configuration in Nagel et al. (2022), we compared Q-EMA & Q-Ramping with their methods. As a result in Tab. 4, their "Dampen" method cannot work well on reducing oscillation

in pre-training, and the "Freezing" method would encounter severe degradation when adapted to pre-training tasks.

**Stability Improvement**  We removed weight quantizers in forward to simulate an oscillation-free training (set $Q^{(1)}$ to identity function), and removed both activation and weight quantizers in forward to simulate a MXFP4 training with stable forward process (set $Q^{(1)}$ and $Q^{(2)}$ to identity function). Consequently, our stabilization method Q-EMA and Q-Ramping can counteract the influence of weight oscillation, and approach a comparable accuracy to training with a full-precision forward process. Results are listed in Tab. 6 in Appendix B.

## 8. Conclusion

In this work, we not only proposed a new MXFP4 training method *TetraJet* for a more accurate 4-bit training in MXFP4 format, but also introduced novel approaches to analyzing and resolving the instability of forward pass, which is the bottleneck of MXFP4 training. Extensive experiments revealed that our *TetraJet* consistently surpasses current 4-bit training methods, and *Q-EMA / Q-Ramping* can provide additional enhancement with effective oscillation reduction, and even achieve competitive performance compared to full-precision training.

## Acknowledgment

The authors gratefully thank Pengle Zhang for the insightful discussions. This work was supported by the NSFC Project (No. 62376131), Beijing Natural Science Foundation (QY24258), Tsinghua University Initiative Scientific Research Program (Student Academic Research Advancement Program), and the High Performance Computing Center, Tsinghua University. J.Z is also supported by the XPlorer Prize.

## Impact Statement

Our MXFP4 low-precision training method enhances AI efficiency, reduces energy consumption, and improves accessibility by lowering hardware costs. This can help bridge technological gaps and promote sustainable AI development. However, the reduced computational cost could lower barriers to malicious uses, such as deepfake generation or automated disinformation. Ensuring that such technologies are used ethically and for the benefit of society is essential to maximizing their positive impact.

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

# A. Statistics for Measuring Oscillation and Training Instability

In this section, we formally define and explain the statistics we use in this paper to measure weight oscillation and training instability.

## A.1. Oscillation Ratio

**Definition**   During a training stage with $T_0$ steps, we sum up updating distance for each master weight element $w$ and its quantized weight $w_Q = Q(w)$:

$$\text{dist}_W = \sum_{t=1}^{T_0} |w^t - w^{t-1}|,$$

$$\text{dist}_Q = \sum_{t=1}^{T_0} |w_Q^t - w_Q^{t-1}|.$$

We define *oscillation ratio* $R_w$ for each weight element, representing the degree of oscillation:

$$R_w := \text{dist}_Q / \text{dist}_W.$$

In the Q-Ramping method for pre-training, we set $T_0 = 30$ to minimize the additional cost of identifying oscillating weights. In the validation experiment (Tab. 6), we set $T_0 = 200$ to fully validate the oscillation reduction.

**Interpretation**   If a weight element $w$ has higher $R_w$ at a certain stage of training, it means that it shows more characteristics of oscillation. The larger $R_w$, the more frequently and severely the weight element $w$ oscillates, which means that we should put more effort into suppressing the oscillation of $w$.

**Comparison of Oscillation Ratio and Previous Metric**   Nagel et al. (2022) also define a metric *flipping frequency $f$* (average frequency of quantization flipping, defined for each weight element) to find out oscillating weights and measure oscillation severity, but it is only suitable for the small learning-rate training (e.g. fine-tuning, or near the end of pre-training), because when the learning-rate is relatively large (e.g. the early or middle stage of pre-training), the latent weight would be updated with large step size and the quantized weights also change frequently during training, but $f$ would *falsely recognize* some of them as quantization oscillation. This is also a reason why the "Freeze" method performs badly in pre-training (see the result in Tab. 4).

Oscillation Ratio $R_w$ overcomes the issue of oscillation detection in the early stage of pre-training. Only the weights that fall into real quantization oscillation would get a large $R_w$: these weights are with small moves around the quantization threshold ($\text{dist}_W$ is relatively small) but with frequent switch between quantization values ($\text{dist}_Q$ is relatively large).

## A.2. Quantization Confidence

**Definition**   To quantitatively assess the severity of the oscillation problem, we define *quantization confidence* for each weight element $w$, which measures the normalized distance to the nearest quantization threshold.

$$\text{QuantConf}(w) := \frac{\min_i |w - \text{thrd}_i|}{\text{MaxDist}(w^{\text{FP4}})}$$

where $w^{\text{FP4}}$ denotes the quantized FP4 value of $w$, $\{\text{thrd}_i\}$ denotes all the quantization thresholds, and $\text{MaxDist}(w^{\text{FP4}})$ denotes the maximum possible distance when quantized to $w^{\text{FP4}}$. It is ensured that $\text{QuantConf}(w) \in [0, 1]$.

**Interpretation**   If an element $w$ has less quantization confidence, it is more prone to oscillate, because it is closer to the quantization threshold and little perturbation would make its quantized value switch frequently. If a weight matrix $\mathbf{W}$ has more elements with low confidence, we call the weight distribution is less confident, which indicates that the optimization to this weight is more unstable. For example, in Fig. 4, weights in Epoch 90 are less confident than weights in Epoch 30.

### A.3. Rate of Change for Weight and Activation

**Definition**  we define *rate of change* for a tensor $\mathbf{X}$ as

$$r(\mathbf{X}) = \frac{1}{T_0} \sum_{t=1}^{T_0} \frac{\left\| \mathbf{X}^t - \mathbf{X}^{t-1} \right\|_F}{\left\| \mathbf{X}^{t-1} \right\|_F}$$

where $t$ refers to training step, and step $0 \sim T_0$ refers to a short training interval.

During pre-training, we can test the rate of change for the master weight $\mathbf{W}$, the quantized weight matrix $Q^{(2)}(\mathbf{W}^\top)^\top$, and activation $\mathbf{Y}$ in different stages.

**Interpretation**  This metric is useful in the end of training. When Learning Rate (LR) is approaching zero to push the model to quickly descend to a local minimum and converge, we expect the rate of change for quantized weight and activation can also be near zero to ensure stability of training. However, in Section 4.1, we have found that the rate of change stays high at the end of MXFP4 training.

Therefore, if we can decrease the rate of change for quantized weight and output activation of quantized layers, it means we effectively improve the training stability. We have shown the results in Tab. 3.

# B. More Detailed Results of Ablation Study

**Quantization Methods**  We do an ablation study to compare our training method TetraJet and Microscaling's original training method. Through Tab. 5, we conclude that: (a) Our *double quantization* corrects the gradient estimation in MXFP4 Linear Layers, and is consistently better than Microscaling's original design. (b) As long as we give **unbiased gradient estimation**, which is guaranteed by *double quantization* and *truncation-free scaling*, we can reach the optimal strategy with *stochastic quantization* in backward. (c) It is necessary to ensure unbiasedness. Only in the unbiased situation, can *stochastic quantization* exert its advantage.

Table 5: Comparison on quantization methods. We report the accuracy on the validation set of 90-epoch DeiT-T pre-training.

| Backward Quant | XW For Grad Computing | Computation of Shared Scale | Top-1% | Top-5% | Note |
|---|---|---|---|---|---|
| **Stochastic** | **Double Quantization** | **Truncation-Free Scaling** | **59.75** | **82.67** | **TetraJet**(unbiased gradient) |
| Stochastic | Double Quantization | Microscaling's Scaling | 59.18 | 82.64 | |
| Stochastic | Microscaling's Design | Truncation-Free Scaling | 56.98 | 80.60 | |
| Stochastic | Microscaling's Design | Microscaling's Scaling | 57.49 | 81.27 | |
| Deterministic | Double Quantization | Truncation-Free Scaling | 58.60 | 82.11 | |
| Deterministic | Double Quantization | Microscaling's Scaling | 59.02 | 82.18 | |
| Deterministic | Microscaling's Design | Truncation-Free Scaling | 58.40 | 81.57 | |
| Deterministic | Microscaling's Design | Microscaling's Scaling | 58.56 | 81.92 | Microscaling |

**Stability Improvement**  We simulated an oscillation-free training by removing the weight quantizer in forward, and simulated a stable forward process by removing both weight & activation quantizers in forward. As a result in Tab. 6, our methods Q-EMA & Q-Ramping can fully eliminate the negative effects of weight oscillation, and can approach better accuracy with a more stable forward process.

**Data Format**  We study the choice of FP4 format for the forward and backward computation. In Tab. 7, although E3M0 is another possible FP4 format, E2M1 is always a better format for weight, activation, and gradient.

Table 6: Ablation study on quantization stability. We report the accuracy on validation set of 90-epoch DeiT-B pre-training. *WQ*: Weight Quantization in forward; *AQ*: Activation Quantization in forward.

|  | Top-1 Acc.% |
|---|---|
| TetraJet | 74.91 |
| TetraJet w/o WQ | 75.16 |
| TetraJet w/o WQ & AQ | 75.86 |
| TetraJet + Q-EMA | 77.32 |
| TetraJet + Q-Ramping | 75.62 |

Table 7: MXFP4 Data Format Selection. We report the top-1 Acc.% of DeiT-T Pre-Training.

| A&W \ Grad | E2M1 | E3M0 |
|---|---|---|
| E2M1 | **59.75** | 58.90 |
| E3M0 | 54.21 | 53.72 |

# C. Detailed Implementation of Q-EMA and Q-Ramping

## C.1. Algorithm: EMA Quantizer (Q-EMA)

---

**Algorithm 1** EMA Quantizer for a Micro-Block (**Q-EMA**)

---

**input** Weight Block $\mathbf{W}$; EMA weight block $\mathbf{W}_{\mathrm{EMA}}$.
**output** Quantized Weight Block $(\mathbf{W}^{\mathrm{FP4}}, s)$ in MXFP4 format
1: Assume $\mathbf{W}$ and $\mathbf{W}_{\mathrm{EMA}}$ are vectors of size 32.
2: $M \leftarrow \max_{1 \leq i \leq 32} |V_i|, \ \widetilde{M} \leftarrow M + \varepsilon \cdot \mathbb{I}(M = 0)$
3: $s \leftarrow \left\lceil \log_2 \frac{2\widetilde{M}}{Q_p - Q_n} \right\rceil, S \leftarrow 2^s$
4: **for** $i \leftarrow 1$ to $32$ **do**
5:      $q_1, q_2 \leftarrow$ two nearest MXFP4 values to $\frac{\mathbf{W}_i}{S}$
6:      **if** $\left| \frac{\mathbf{W}_{\mathrm{EMA}i}}{S} - q_1 \right| < \left| \frac{\mathbf{W}_{\mathrm{EMA}i}}{S} - q_2 \right|$ **then**
7:         $\mathbf{W}_i^{\mathrm{FP4}} \leftarrow q_1$
8:      **else**
9:         $\mathbf{W}_i^{\mathrm{FP4}} \leftarrow q_2$
10:      **end if**
11: **end for**
12: Return MXFP4 block $(\mathbf{W}^{\mathrm{FP4}}, s)$

---

## C.2. Algorithm: Adaptive Ramping Optimizer (Q-Ramping)

---

**Algorithm 2** Adaptive Ramping Algorithm for MXFP4 Training (**Q-Ramping**)

---

1: **Hyperparameter**: $k_1, k_2$.
2: **function** OscillationDetection(Model $M$, Global Learning-Rate LR, Global Batch-Size BS)
3:      Train the model $M$ for $T_0 \ll T_{\mathrm{update}}$ steps on a calibration dataset *without Q-Ramping*, to detect oscillating weight.
4:      **for** each weight element $w$ **in** quantized layers **do**
5:         Compute the *oscillation ratio* $R_w$ according the length of trajectory of master weight $w$ & quantized weight $w_Q$;
6:         $\mathrm{LR}_w \leftarrow \min(k_2 \lfloor R_w / k_1 \rfloor + 1, N_{\max}) \cdot \mathrm{LR}$;
7:         $\mathrm{BS}_w \leftarrow \min(k_2 \lfloor R_w / k_1 \rfloor + 1, N_{\max}) \cdot \mathrm{BS}$;
8:         // $k_1, k_2$ are coefficients for amplifying LR & BS (that is using a higher gradient accumulation step).
9:         // $N_{\max}$ denotes the maximum amplification factor.
10:      **end for**
11: **end function**
12: **function** ModelTraining_with_Q-Ramping(Initial Model $M$, Steps $T$, Learning-Rate LR, Batch-Size BS)
13:      **for** $t \leftarrow 0$ to $T$ **do**
14:         **if** $t \bmod T_{\mathrm{update}} = 0$ **then**
15:            call OscillationDetection($M$, LR, BS) to adaptively adjust $\mathrm{LR}_w$ & $\mathrm{BS}_w$ for each element $w$;
16:         **end if**
17:         **for** each weight element $w$ **in** quantized layers **do**
18:            update $w$ according to $\mathrm{LR}_w$ & $\mathrm{BS}_w$ by Customized AdamW;
19:         **end for**
20:         **for** each parameter $\mathbf{W}$ **in** non-quantized layers **do**
21:            update $\mathbf{W}$ by normal AdamW;
22:         **end for**
23:      **end for**
24: **end function**

---

## C.3. Selection of Hyperparameter & Insensitity to Hyperparameter

For Q-EMA, the momentum $\beta = 0.998$ for calculating $\mathbf{W}_{\text{EMA}}$ is a good default choice. For Q-Ramping, $k_1 = 16$ is a good threshold to measure the severity of oscillation, and $k_2 = 5$ is a default ratio for amplifying the Learning Rate & Batch Size (meanwhile, reducing the frequency of oscillation). We can reach better performance through minor tuning. The detailed settings are listed in Tab. 8.

Table 8: Selection of hyperparameter in Q-EMA & Q-Ramping.

|  | DeiT-T | DeiT-S | DeiT-B | Swin-T | Swin-S |
|---|---|---|---|---|---|
| TetraJet | 59.75 | 71.03 | 74.91 | 77.12 | 79.51 |
| TetraJet + Q-EMA (default: $\beta = 0.998$) | 59.69 | 71.51 | 77.18 | 77.23 | 79.74 |
| TetraJet + Q-EMA (best: $\beta$ tuned) | 60.00 ($\beta = 0.9983$) | 72.25 ($\beta = 0.9972$) | 77.32 ($\beta = 0.999$) | 77.30 ($\beta = 0.9975$) | 79.74 ($\beta = 0.998$) |
| TetraJet + Q-Ramping (default: $k_1 = 16, k_2 = 5$) | 60.31 | 71.32 | 75.62 | 77.23 | 79.52 |
| TetraJet + Q-Ramping (best: $k_1 = 16$, $k_2$ tuned) | 60.31 ($k_2 = 5$) | 71.32 ($k_2 = 5$) | 75.62 ($k_2 = 5$) | 77.33 ($k_2 = 3$) | 79.67 ($k_2 = 4$) |

We also validate Q-EMA / Q-Ramping's insensitivity to hyperparameter choice in Tab. 9 & 10.

Table 9: Insensitivity to hyperparameters (TetraJet + Q-EMA) on DeiT-B.

| $\beta$ | 0.993 | 0.995 | 0.997 | 0.998 | 0.999 | 0.9995 | w/o Q-EMA |
|---|---|---|---|---|---|---|---|
| Accuracy | 75.39 | 76.37 | 77.23 | 77.18 | **77.32** | 77.30 | 74.91 |

Table 10: Insensitivity to hyperparameters (TetraJet + Q-Ramping) on DeiT-B.

| $k_1$ | 16 | 16 | 16 | 16 | 16 | 16 | 8 | 12 | 16 | 20 | w/o Q-Ramping |
|---|---|---|---|---|---|---|---|---|---|---|---|
| $k_2$ | 3 | 4 | 5 | 6 | 7 | 8 | 5 | 5 | 5 | 5 | |
| Accuracy | 75.35 | 75.33 | **75.62** | 74.96 | 75.29 | 75.13 | 75.19 | 75.60 | **75.62** | 74.85 | 74.91 |

## C.4. Other Discussion on Q-EMA & Q-Ramping

Q: Why cannot we combine two algorithms?

A: When we use Q-EMA, there are two variables ($\mathbf{W}$ & $\mathbf{W}_{\text{EMA}}$) that determine the result of weight quantization, so training with Q-EMA results in a different MXFP4 training dynamic. Therefore, it is more complicated to identify and track the oscillating weights in this situation. Therefore, it is not proper to simply combine Q-EMA & Ramping.

