# OpenReview forum: "Oscillation-Reduced MXFP4 Training for Vision Transformers"
_ICML.cc/2025/Conference — ICML 2025 poster_

### Official Review · Reviewer_TZFB · 2025-03-03

**Overall Recommendation:** 3

**Summary:**

This paper proposes two methods to train vision transformers with MXFP4-accelerated GEMMs. In the backward pass, the authors add stochastic rounding and scaling to achieve unbiased gradient estimates. In the forward pass, the authors add various EMA-based methods to avoid weight oscillation during quantization. The authors test their method on training smaller vision transformers and find that the proposed methods are sufficient to achieving near-lossless performance.

**Claims And Evidence:**

Yes

**Essential References Not Discussed:**

A number of FP4 LLM training papers have come out since this paper was submitted. The authors may find these works useful and interesting but since they were released around the time of submission or after, it would be unreasonable to have the authors compare against them.

**Experimental Designs Or Analyses:**

The experiments seem reasonable.

**Methods And Evaluation Criteria:**

Yes

**Other Comments Or Suggestions:**

Is double quantization actually necessary? If you use stochastic rounding to compute unbiased gradients and stochastic rounding is implemented with iid noise, the output should be unbiased regardless of which ways the MX groups go. Perhaps I am misunderstanding what the core issue is that necessitates double quantization.

Is there a reason you chose to evaluate on small vision transformers? This method does not seem specific to vision transformers and could be applied to language models as well.

What is the cost of storing the EMA components for oscillation reduction?

**Other Strengths And Weaknesses:**

The paper seems to have two separate methods: unbiased gradient estimation in the backward pass and oscillation reduction in the forward pass. It is not clear to me how orthogonal these two methods are. Are unbiased gradients necessary to get oscillation reduction to work, or is oscillation reduction necessary to get unbiased gradients to have an effect on the training curve?

**Questions For Authors:**

See above.

**Relation To Broader Scientific Literature:**

FP4 training is a relatively unsolved problem. The paper makes a number of references to oscillation reduction in low precision training that seem relevant.

**Theoretical Claims:**

This is a mostly empirical paper.

---

> ### Author Rebuttal · Authors · 2025-03-31
>
> We thank the reviewer for valuable comments and the acknowledgment of our contributions. Below we respond to the questions.
> > Question 1:
> > … Are unbiased gradients necessary to get oscillation reduction to work, or is oscillation reduction necessary to get unbiased gradients to have an effect on the training curve?
>
> Thanks for the valuable questions. In short, the methods for forward (oscillation-reduction) and for backward (unbiased gradients) are **orthogonal**. We provide empirical evidence as follows.
>
> For the first question, we conduct an additional experiment to show that oscillation reduction is not limited to a certain method of gradient calculation (Table 1). We can see the Q-EMA also works on the baseline Microscaling (MS), which does not unbiasedly estimate gradients.
>
> For the second question, we have done experiments in our work without oscillation reduction techniques. In Table 2 (part of Table 2 in our paper), our method TetraJet(TJ) with quantization improvement on unbiased estimation (do not add oscillation reduction technique) outperforms baseline Microscaling (MS).
>
> *Table 1. Oscillation Reduction on DeiT-S 60-epoch pre-training.*
> | Methods | Accuracy |
> | --- | --- |
> | MS (Baseline) | 63.73 |
> | MS+Q-EMA | 64.19 |
>
> *Table 2. Accuracy improvement without additional oscillation reduction techniques (90-epoch pre-train)*
> |Methods|DeiT-T|DeiT-S|DeiT-B|Swin-T|Swin-S|
> |-|-|-|-|-|-|
> |MS(Baseline)|58.56|70.10|74.54|76.87|79.45|
> |TJ(Ours)|59.75|71.03|74.91|77.12|79.51|
>
> > Question 2:
> > Is double quantization actually necessary?
>
> Thanks for the valuable question. There are mainly two core issues that necessitate double quantization：
> - **(1) The correctness of optimization goal**. The forward pass of our optimized network is $Y= Q_D^{(1)}(X) \times Q_D^{(2)}(W^\top)$. So the gradient for weight is $\nabla_{W}\mathcal{L}\overset{\text{STE}}{\approx} \nabla_{Q_D^{(2)}(W)} \mathcal L = (\nabla_Y\mathcal L)^\top \times Q_D^{(1)}(X)$. To unbiasedly calculate this gradient using MXFP4, we need to estimate $Q_D^{(1)}(X)$ rather than $X$, to align with the forward pass.
> - **(2) The non-square group shape**. To enable efficient MXFP4 Matrix Multiplication, the group shapes should be (1x32) x (32x1) for each multiplication. Because $Q_D^{(1)}(X)$ is in group shape 1x32 in forward, we need another unbiased quantizer for $Q_D^{(1)}(X)$ to achieve group shape 32x1 during backward, so we get the double-quantized $Q_S^{(6)}\left(Q_D^{(1)}(X)\right)$ (see Eq.5 of our paper).
>
> In conclusion, the correctness of optimization goal & the non-square group shape (1x32 / 32x1) make double quantization necessary in our MXFP4 training.
>
> > Question 3:
> > Is there a reason you chose to evaluate on small vision transformers? This method does not seem specific to vision transformers and could be applied to language models as well.
>
> Thanks for the valuable comment. We chose to begin our exploration of FP4 training with Vision Transformers (ViTs) primarily due to computational resource constraints, as pre-training large language models (LLMs) in FP4 is significantly more resource-intensive. Importantly, developing efficient FP4 training algorithms for vision tasks is itself a meaningful and underexplored direction, with many prior works also focusing on ViTs (e.g., [1,2,3]).
>
> Our ultimate goal is to enable FP4 training for LLMs, and we actually observe that our proposed method, TetraJet, generalizes well to LLMs and outperforms FP4 baselines (see Table 3). We also find that oscillation problem still exists in LLM pertaining task. However, fully resolving convergence issues and matching BF16 performance in large-scale LLMs still requires further systematic investigation, particularly due to their large scale and dynamics. We believe that our work on ViTs Training with FP4 formats would be inspiring and valuable for further exploration of LLMs training.
>
> *Table 3. OLMo-2 150M pre-training with 20 Billion tokens*
> |Method|Perplexity|
> |-|-|
> |BF16  | 23.18  |
> |Microscaling(Baseline)|25.88|
> |TetraJet(Ours)|23.82|
>
> > Question 4:
> > What is the cost of storing the EMA components for oscillation reduction?
>
> During training, we only need to store an extra EMA weight for each linear layer in transformer blocks. For DeiT-B with 85M linear parameters in Transformer blocks, this adds about 340MB of storage, which is relatively small compared to the overall training memory footprint (around 20GB per GPU in a 4-GPU setup).
>
> If we use FP4-trained models for inference, we can calculate FP4 values for each parameter in advance, so we don’t need the EMA component anymore during inference, which means no additional cost.
>
> [1] Y. Li et al.,"Q-vit: Accurate and fully quantized low-bit vision transformer," NeurIPS, 2022.
> [2] Y. Liu et al.,"NoisyQuant: Noisy bias-enhanced post-training activation quantization for vision transformers," CVPR, 2023.
> [3] Z. Wang et al.,"Quantformer: Learning extremely low-precision vision transformers," IEEE TPAMI,2022.

---

> > ### Comment · Reviewer_TZFB · 2025-04-04
> >
> > Thank you for your response and additional experiments. I will keep my score.

---

### Official Review · Reviewer_KRL2 · 2025-03-09

**Overall Recommendation:** 4

**Summary:**

The authors propose a MXF4 training scheme for Vision Transformers. Training at extremely low-bit such as 4-bit formats is challenging and prone to high accuracy loss, mainly due to weight oscillations in the forward pass as identified by the authors. The paper outlines two methods EMA Quantizer (Q-EMA) and Adaptive Ramping Optimizer (Q-Ramping) to resolve this issue.

**Claims And Evidence:**

The authors provide clear evidence through detailed experiments on popular Vision Transformer architectures (DeiT and Swin). They systematically identify the accuracy degradation problem in MXFP4 training due to weight oscillations. The evidence from the ablation studies is thorough, clearly indicating that forward-pass quantizers for activations and weights cause the most degradation. Both proposed solutions, Q-EMA and Q-Ramping, effectively address this oscillation issue, demonstrated through quantitative metrics like rate of change and quantization confidence.
The experiments and analyses are solid, with clear ablation studies and comparisons against baseline methods and competitive state-of-the-art methods. I reviewed the supplementary materials

**Essential References Not Discussed:**

NA

**Experimental Designs Or Analyses:**

Results and analysis sound reasonable.

**Methods And Evaluation Criteria:**

The proposed method sound reasonable with proper explanation.

**Other Comments Or Suggestions:**

I think the sentence "Therefore, quantizer (1)(3)(5) should use 1 × 32 group shape, and quantizer (2)(4)(6) should use
32 × 1 group shape." can be better phrased. It's hard to follow the sentence and what quantizer(*) refer to.

**Other Strengths And Weaknesses:**

Strengths:

- Clearly identifies and addresses a practical issue (weight oscillation) with thoughtful solutions.
- Extensive empirical validation across multiple ViT architectures demonstrates significant performance improvements.
- Methods introduced (Q-EMA, Q-Ramping) are innovative yet practical.

Weaknesses:

- The analysis primarily focuses on ViT architectures; evaluating more diverse architectures (e.g., CNNs, LLMs) could further validate general applicability.
- Discussion of potential overhead or computational costs associated with the oscillation tracking in Q-Ramping could be expanded.

**Questions For Authors:**

- Could you clarify the computational overhead introduced by Q-EMA and Q-Ramping methods in actual training?
- Are the proposed methods sensitive to network architecture or hyperparameters beyond those tested? Specifically, how robust are they for larger models beyond the evaluated ViTs?

**Relation To Broader Scientific Literature:**

The paper tackles the complex problem of training with MXFP4 format. There are several recent attempts at low-bit training specifically targeting FP8, and INT4 methods. The discussion of prior techniques (e.g., Microscaling, Jetfire, LSQ quantization) is comprehensive and clear. However, the authors might enhance their context by explicitly citing some recent low-precision training surveys, if available, to provide broader context.

**Theoretical Claims:**

I did not rigorously verify any theoretical proofs, as the primary contributions are experimental and methodological.

---

> ### Author Rebuttal · Authors · 2025-03-31
>
> We thank the reviewer for valuable comments and the acknowledgment of our contributions. Below we respond to the questions.
>
> > Broader Scientific Literature:
> > The authors might enhance their context by explicitly citing some recent low-precision training surveys...
>
> Thanks for the valuable advice. We have found some surveys on low-precision training [1,2,3]. We will include them in our revised version.
>
> > Other Comments Or Suggestions:
> > I think the sentence "Therefore, quantizer …. group shape." can be better phrased. It's hard to follow the sentence and what quantizer(*) refer to.
>
> Thanks for the valuable advice. We will clarify this by pointing out that they are Quantizers $Q^{(*)}$ in Eq. 3-5.
>
> > Weakness 1:
> > The analysis primarily focuses on ViT architectures; evaluating more diverse architectures (e.g., CNNs, LLMs) could further validate general applicability.
>
> Thanks for the valuable comments. According to NVIDIA's design[4], block scaling formats (e.g., MXFP4) are mostly designed for matrix multiplications rather than convolutions. Therefore, our work focuses on designing methods tailored to Transformers rather than CNNs.
>
> Due to space limitations, we left the detailed response to this question in the rebuttal to “Question 3” of Reviewer TZFB. In short, we include results demonstrating that our method generalizes to LLMs, suggesting that our approach can be a promising direction for FP4 training of LLMs as well.
>
> > Weakness 2:
> > Discussion of potential overhead or computational costs associated with the oscillation tracking in Q-Ramping could be expanded.
>
> Thanks for the constructive comment. In Q-Ramping method, we do not track oscillations all the time. For example, in ImageNet-1K pre-training, an epoch consists of ~2500 iterations. We only track the first 30 iterations to find the oscillating weights. As a result, this only adds 1.64% to the total training wall time compared to w/o Q-Ramping in our DeiT-B training on RTX4090.
>
> > Question 1:
> > Could you clarify the computational overhead introduced by Q-EMA and Q-Ramping methods in actual training?
>
> Thanks for the valuable questions.
> In conclusion, both methods only introduce little overhead into the training time. For Q-EMA method, we only need to use EMA for weight quantization during the forward pass (other quantizers don’t have additional cost), and we update the EMA-weight for quantized layers when parameters are updated. In our DeiT-B training on RTX4090, we found Q-EMA only adds 0.35% to the total training wall time compared to w/o Q-EMA. For Q-Ramping method, we have discussed above in response to Weakness 2.
>
>
> > Question 2:
> > Are the proposed methods sensitive to network architecture or hyperparameters beyond those tested? Specifically, how robust are they for larger models beyond the evaluated ViTs?
>
> Thanks for the valuable questions. We have evaluated the robustness of hyperparameters for DeiT-B. As Tables 1&2 show, Q-EMA’s decay factor β and Q-Ramping’s update frequency factors $k_1$, $k_2$ are not sensitive within certain intervals.
>
> We conducted additional fine-tuning experiments for ViT-base model [6] using the same default hyperparameters from our paper, and found our methods are not sensitive to network architectures or specific tasks (Table 3). Furthermore, our methods significantly outperform the baseline in LLMs pre-training using the larger OLMo-2-150M (Table 4), showing generalization beyond ViT models. These results demonstrate that our method can be adopted to different kinds of architectures and tasks.
>
> *Table 1: Insensitivity to hyperparameters (Q-EMA) on DeiT-B.*
>
> | $\beta$ | 0.993 | 0.995 | 0.997 | 0.998 | 0.999 | 0.9995 | w/o Q-EMA |
> | ------- | ----- | ----- | ----- | ----- | ----- | ------ | --------- |
> |Acc%|75.39|76.37|77.23|77.18|77.32|77.30|74.91|
>
> *Table 2: Insensitivity to hyperparameters (Q-Ramping) on DeiT-B.*
> |$k_1$|16|16|16|16|16|16|8|12|16|20|w/o Q-Ramping|
> |-|-|-|-|-|-|-|-|-|-|-|-|
> |$k_2$|3|4|5|6|7|8|5|5|5|5||
> |Acc%|75.35|75.33|75.62|74.96|75.29|75.13|75.19|75.60|75.62|74.85|74.91|
>
> *Table 3. Results of 50epoch MAE ViT-base Fine-tuning (MS: Microscaling; TJ: TetraJet)*
> |Methods|Acc%(mean±std)|
> |---|---|
> |BF16|81.49±0.08|
> |MS(Baseline)|80.04±0.04|
> |TJ(Ours)|80.17±0.03|
> |TJ+Q-EMA(Ours)|80.57±0.09|
> |TJ+Q-Ramping(Ours)|80.25±0.04|
>
>
> *Table 4. OLMo-2 [6] 150M Pre-training with 20 Billion tokens*
> | Methods | Perplexity |
> | --- | --- |
> | BF16  | 23.18  |
> | MS (Baseline) | 25.88  |
> | TJ (Ours)| 23.82 |
>
> [1] Wei L et al., "Advances in the Neural Network Quantization: A Comprehensive Review". Applied Sciences. 2024.
> [2] Chitsaz K et al., "Exploring Quantization for Efficient Pre-Training of Transformer Language Models," arXiv:2407.11722, 2024.
> [3] Kumar T et al., "Scaling Laws for Precision," arXiv:2411.04330, 2024.
> [4] https://docs.nvidia.com/cuda/pdf/ptx_isa_8.7.pdf
> [5] OLMo Team et al., "2 OLMo 2 Furious," arXiv:2501.00656, 2024.
> [6] https://github.com/facebookresearch/mae

---

### Official Review · Reviewer_e9Me · 2025-03-14

**Overall Recommendation:** 3

**Summary:**

The paper introduces TetraJet, a novel training method for Vision Transformers using the MXFP4 low-precision format, which is supported by Nvidia's Blackwell GPUs and offers significant speed improvements. The authors identify weight oscillation as a key issue causing accuracy degradation in MXFP4 training. To address this, they propose two techniques: EMA Quantizer (Q-EMA), which smooths weight quantization using an exponential moving average, and Adaptive Ramping Optimizer (Q-Ramping), which dynamically reduces update frequency for oscillating weights. Their approach achieves over 50% reduction in accuracy degradation compared to the baseline Microscaling method and brings MXFP4 training close to full-precision performance, demonstrating its effectiveness in stabilizing low-precision training.

**Claims And Evidence:**

The paper provides strong empirical evidence to support its claims, including extensive experiments on Vision Transformers that demonstrate TetraJet’s superiority over existing MXFP4 training methods. The identification of the weight oscillation problem is backed by quantitative analysis of weight changes, rate of change metrics, and oscillation ratio measurements. The effectiveness of Q-EMA and Q-Ramping is validated through comparative accuracy results, stability improvements, and confidence metrics, showing clear advantages over the baseline. However, while the paper convincingly argues that oscillation is the main source of degradation, it does not theoretically prove why these methods generalize across different architectures or tasks beyond Vision Transformers, leaving room for further validation.

**Essential References Not Discussed:**

None

**Experimental Designs Or Analyses:**

The experimental design is robust and well-structured, leveraging ImageNet-1K as a benchmark and conducting comprehensive ablation studies to isolate the effects of different quantization components. The impact analysis of six quantizers (Table 1) effectively identifies the activation and weight quantizers in the forward pass as the primary sources of degradation. The oscillation reduction experiments (Figures 2–6) provide strong empirical support for Q-EMA and Q-Ramping. However, the study lacks statistical significance testing (e.g., confidence intervals or variance analysis), which would further strengthen the validity of the reported improvements. Additionally, while Q-Ramping's dynamic adaptation is intuitive, its hyperparameter sensitivity is only briefly discussed, and a more detailed analysis of its tuning could enhance confidence in its robustness.

**Methods And Evaluation Criteria:**

The methods and evaluation criteria are well-aligned with the problem of low-precision training for Vision Transformers. The authors conduct experiments on ImageNet-1K using established Vision Transformer architectures (DeiT and Swin Transformers), which are widely used benchmarks for image classification. Their evaluation focuses on accuracy degradation, oscillation metrics, and training stability, which are appropriate for assessing the impact of quantization techniques. The use of quantization confidence and oscillation ratio provides insightful analysis beyond standard accuracy metrics. However, the paper primarily evaluates MXFP4 on pre-training tasks, and it would be useful to see results on fine-tuning or downstream applications to confirm generalizability.

**Other Comments Or Suggestions:**

None

**Other Strengths And Weaknesses:**

- The study focuses only on Vision Transformers and does not evaluate other architectures (e.g., CNNs, NLP models like LLMs), limiting its broader applicability.

**Questions For Authors:**

- Have you tested MXFP4 fine-tuning on pre-trained models? Does the oscillation issue persist in fine-tuning settings?
- How sensitive are Q-EMA’s decay factor (β) and Q-Ramping’s update frequency adjustments to different datasets and architectures? Did you find any optimal ranges?

**Relation To Broader Scientific Literature:**

The paper builds on prior work in low-precision training, QAT, and MX quantization (Rouhani et al., 2023) by addressing oscillation issues in MXFP4 training from scratch. It extends findings on weight oscillation in QAT (Nagel et al., 2022; Liu et al., 2023) but uniquely applies EMA-based smoothing and adaptive update strategies to stabilize MXFP4. While related techniques exist in optimization, their use for 4-bit Vision Transformer training is novel. However, its applicability beyond vision models remains unexplored.

**Theoretical Claims:**

The paper primarily focuses on empirical findings, but it includes some theoretical justifications for its quantization techniques, such as the unbiased gradient estimation from double quantization and truncation-free scaling. The derivations in Section 3.4 align with prior work on Straight-Through Estimators (STE) and unbiased gradient estimation. The quantization confidence metric and the oscillation ratio definition are intuitively reasonable, though they lack formal theoretical validation. While the explanations are convincing, rigorous mathematical proofs on why Q-EMA and Q-Ramping reduce oscillation across different scenarios are not provided, leaving some theoretical gaps.

---

> ### Author Rebuttal · Authors · 2025-03-31
>
> We thank the reviewer for valuable comments. Below, we respond to the questions.
>
> > Claims And Evidence & Theoretical Claims:
> > Lack formal theoretical validation & Leaving some theoretical gaps
>
> Thank you for the insightful comment. We agree that theoretical analysis is important. While our current work primarily focuses on empirical evaluation, we acknowledge the need for a stronger theoretical foundation to support our findings on oscillation reduction. However, theoretically analyzing the convergence properties of low-precision Transformer training remains an open and challenging problem, primarily due to the non-differentiability of quantization functions. We view this as a valuable direction for future research and are actively exploring ways to address it.
>
> > Other Weaknesses:
> > The study focuses only on Vision Transformers and does not evaluate other architectures (e.g., CNNs, NLP models like LLMs).
>
> Thanks for the valuable comment. According to NVIDIA's design[1], block scaling formats (e.g., MXFP4) are mostly designed for matrix multiplications rather than convolutions. Therefore, our work focuses on designing methods tailored to Transformers rather than CNNs.
>
> Due to space limitations, we left the detailed response to this question to the rebuttal to “Question 3” of Reviewer TZFB. In short, we include results demonstrating that our method generalizes to LLMs, suggesting that our approach can be a promising direction for FP4 training of LLMs as well.
>
> > Experimental Designs Or Analyses:
> > the study lacks statistical significance testing (e.g., confidence intervals or variance analysis), which would further strengthen the validity of the reported improvements.
>
> Thanks for the valuable advice and questions. We report results with standard deviation based on three runs using different random seeds in our additional fine-tuning experiments (see Table 1). Results demonstrating that our method performs consistently better than baselines.
>
> > Question 1:
> >Have you tested MXFP4 fine-tuning on pre-trained models? Does the oscillation issue persist in fine-tuning settings?
>
> Our research is mainly focused on ViTs pre-training, but can also be generalized to fine-tuning. Our further conducted experiments show that the oscillation problem still exists, which aligns with previous literature on oscillation problem about low-precision fine-tuning [2].
>
> We finetune MAE-ViT-base [3] for 50 epochs based on the pre-trained model, and we report the average accuracy and standard deviation over 3 seeds. Our method TetraJet still outperforms the baseline, and Q-EMA/Q-Ramping provides additional enhancement by alleviating the oscillation problem.
>
> *Table 1. Results of ViT-base Fine-tuning (MS: Microscaling; TJ: TetraJet)*
> | Methods | Acc% (mean±std) |
> | --- | ---|
> | BF16 | 81.49±0.08 |
> | MS (Baseline) | 80.04±0.04|
> | TJ (Ours) | 80.17±0.03 |
> | TJ + Q-EMA (Ours)  | 80.57±0.09 |
> | TJ + Q-Ramping (Ours) | 80.25±0.04 |
>
> > Question 2:
> > How sensitive are Q-EMA’s decay factor (β) and Q-Ramping’s update frequency adjustments to different datasets and architectures? Did you find any optimal ranges?
>
> Thanks for the valuable questions. The choices of these hyperparameters are **not sensitive** through our experiments. As Table 2&3 show, Q-EMA’s decay factor β and Q-Ramping’s update frequency factors $k_1$, $k_2$ are not sensitive within certain intervals.
> We find [0.997, 0.999] is an optimal range for Q-EMA’s decay factor (β); a good choice for oscillation detection factor $k_1$ is 16, and good weight update frequency factor $k_2$ can be from 3 to 5.
>
> We used default settings ($\beta=0.998, k_1=16,k_2=5$) for our reported pre-training and fine-tuning experiments, and found additional enhancement of Q-EMA and Q-Ramping across different architectures and tasks (DeiT/Swin pre-training & ViT fine-tuning). These results further confirmed the robustness and the ability of generalization of our oscillation-reduction methods.
>
> *Table 2: Insensitivity to hyperparameters (TetraJet + Q-EMA) on DeiT-B.*
>
> | $\beta$ | 0.993 | 0.995 | 0.997 | 0.998 | 0.999 | 0.9995 | w/o Q-EMA |
> | ------- | ----- | ----- | ----- | ----- | ----- | ------ | --------- |
> | Acc%    | 75.39 | 76.37 | 77.23 | 77.18 | 77.32 | 77.30  | 74.91     |
>
>
> *Table 3: Insensitivity to hyperparameters (TetraJet + Q-Ramping) on DeiT-B.*
> | $k_1$ |  16   |  16   |  16   |  16   |  16   |  16   |   8   |  12   |  16   |  20   | w/o Q-Ramping |
> | --- | --- | --- | --- | --- | --- | --- | --- | --- | --- | --- | --- |
> | $k_2$ |   3   |   4   |   5   |   6   |   7   |   8   |   5   |   5   |   5   |   5   |               |
> |  Acc  | 75.35 | 75.33 | 75.62 | 74.96 | 75.29 | 75.13 | 75.19 | 75.60 | 75.62 | 74.85 |     74.91     |
>
> [1] https://docs.nvidia.com/cuda/pdf/ptx_isa_8.7.pdf
> [2] S.-Y. Liu, Z. Liu, and K.-T. Cheng, "Oscillation-free quantization for low-bit vision transformers," ICML, PMLR, 2023.
> [3] https://github.com/facebookresearch/mae

---

> > ### Comment · Reviewer_e9Me · 2025-04-07
> >
> > Thank you for your prompt reply and for the additional experiments. I will maintain my original score.

---

### Decision · Program_Chairs · 2025-05-01

**Decision:**

Accept (poster)

**Comment:**

Summary: The paper introduces TetraJet training method for Vision Transformers using MXFP4 low-precision data type. The paper investigates the root cause of degradation in performance and identify weight oscillation as a root cause. The paper further fixes this using EMA method which smoothens weight quantization and Q-Ramping which reduces updates for unstable weights. The paper demonstrates experimental results via training DeiT and Swin transformers.

Review summary:
Reviewers agree this is an important paper for low-precision training and found the paper is well written with good empirical results. The proposed solutions are simple, effective, and supported by ablation studies. Overall, reviewers are satisfied and lean toward acceptance.